# Outbreak of Alimentary Tick-Borne Encephalitis in Eastern Slovakia: An Analysis of Affected Patients and Long-Term Outcomes

**DOI:** 10.3390/pathogens11040433

**Published:** 2022-04-02

**Authors:** Zuzana Paraličová, Jakub Sekula, Pavol Jarčuška, Martin Novotný, Alena Rovňáková, Ján Hockicko, Ivana Hockicková

**Affiliations:** Department of Infectology and Travel Medicine, Faculty of Medicine, Louis Pasteur University Hospital, Pavol Jozef Šafárik University, 041 90 Kosice, Slovakia; zuzana.paralicova@upjs.sk (Z.P.); jakub.sekula@unlp.sk (J.S.); jarcuska@gmail.com (P.J.); martin.novotny@unlp.sk (M.N.); alena.rovnakova@unlp.sk (A.R.); jan.hockicko@unlp.sk (J.H.)

**Keywords:** tick-borne encephalitis, alimentary infection, clinical manifestation, consequences, Slovakia

## Abstract

Objective: Tick-borne encephalitis (TBE) is an endemic zoonotic viral disease in many European countries and in the central and eastern parts of Asia. Slovakia reports the highest occurrence of alimentary tick-borne encephalitis in Europe, after the consumption of unpasteurized milk and cheese from domestic ruminants. In May 2016, an outbreak of tick-borne encephalitis that emerged after the consumption of sheep cheese occurred in eastern Slovakia. In total, 44 people were ill and 36 were hospitalized. Methods: Data from the 36 hospitalized patients at the Department of Infectology and Travel Medicine in Košice with TBE were retrospectively analysed from the medical documentation. The patients were contacted 3 years after discharge. Results: Twenty of the hospitalized patients had meningoencephalitis and 16 had meningitis. The main symptoms that occurred in all patients were fever and headache. Nuchal rigidity was seen in 50% of the patients. Three patients developed late systemic complications and another six patients had psychiatric complications. None of the patients died. Three years after the disease onset, 52% of contacted patients reported persistent discomfort. Conclusions: TBE is an infection with a wide range of clinical courses. Our findings suggest that alimentary-acquired TBE lead to severe disease and persistent discomfort.

## 1. Introduction

Tick-borne encephalitis (TBE) is a disease that was first described in the first half of the 20th century in Austria, and later, a more severe form in Russia. In the last two decades, the incidence of TBE has increased, as well as its distribution to new territories. One of the causes of this development is climate change, which is leading to a greater geographic distribution of *Ixodes* ticks [1,2]. The expansion of ticks into green urban areas in Europe has also been observed [3]. Currently, TBE occurs from Western Europe to Russia, and further reaches Japan and China. The disease is caused by a TBE virus from the family *Flaviviridae*, genus *Flavivirus*. Currently, there are three established subtypes, with differing geographic distributions, even though there is evidence of their overlap in the different territories: 1. Far Eastern (formerly known as Russian Spring–Summer Encephalitis), 2. Siberian (formerly West-Siberian) and 3. European–Western type (formerly known as the Central European type). Two new candidate subtypes (Himalayan and Baikalian) have been recently discovered [4,5]. 

The disease is transmitted by ticks. The main vector is *Ixodes ricinus* (European subtype) [6] and *Ixodes persulcatus* (eastern subtypes) [7,8]. Other tick species, such as *Dermacentor reticulatus*, *Ixodes gibbosus,* may also be occasional vectors [6,9,10].

The infection may even occur after the ingestion of unpasteurized cow, goat or sheep milk or their products. Whilst tick-borne cases are sporadic, alimentary infections usually involve smaller or larger outbreaks.

The course of the disease is, in most cases, subclinical. The disease is more likely to be clinically manifested in elderly people, with more frequent neurological manifestations [11].

In May 2016, an outbreak of tick-borne encephalitis occurred in the Košice region of eastern Slovakia, caused by the consumption of unpasteurized sheep cheese. The cheese was made at a chalet in the village of Košická Belá, which belongs to the agricultural farm of the village Nižný Klátov.

The aim of this study was to analyse epidemiological data, clinical manifestations and long-term outcomes in patients hospitalized for TBE, in connection with this epidemic and the potential risk factors predisposing them to long-term consequences of the disease.

## 2. Results

### 2.1. Epidemiological Data

All patients in the group declared consumption of unpasteurized sheep cheese from the farm in Nižný Klátov; only one reported four tick bites one month before the illness. The patients reported consuming the cheese purchased between 29 April 2016 and 15 May 2016, directly from the agricultural farm. Through epidemiological inquiry, the number of exposed people was estimated to be 500, but they were not examined for the presence of anti-TBEV antibodies [12]. 

None of the patients had been previously vaccinated against TBE, and none admitted previous *Flavivirus* infection. The mean incubation period from the consumption of the cheese to symptom onset was 14 days (min. 4 days, max. 28 days).

### 2.2. Clinical Course

Twenty-five (69%) patients had biphasic clinical course. The typical symptoms of neuroinfection are headache, fever, nuchal rigidity, postural instability, nausea and vomiting, photophobia and confusion. The constant symptoms that occurred in all hospitalised patients with acute TBE were fever and headache. Positive meningeal signs, such as nuchal rigidity, were seen in only half of the patients. The incidence of individual symptoms in the patient population is shown in Table 1.

Twenty patients had severe disease, characterized as meningoencephalitis with neurological impairment. The most common neurological symptoms were: upper limb tremor (13×), altered level of consciousness—confusion (6×), vertigo (3×), palsy of the lower extremities (1× paraparesis, 1× paresis of right leg), cerebellar syndrome (2×), memory dysfunction (2×) and insomnia (2×). There were also single cases of diplopia, aphasia, hyperreflexia, tetany, fine motor disorder and paraesthesia reported. Sixteen patients had meningitis with no focal neurological deficits; two of them had a cough, one had increased activity of the liver enzymes and one had a fatigue syndrome. The average length of hospitalization was 7 days (ranging from 4 to 12 days). None of the patients died. Neurological symptoms, such as palsy of the lower extremity, fine motor disorder, tremor, insomnia and memory deficits, persisted with less intensity, even after patients were released. During regular check-ups, these symptoms disappeared over time, but one 49-year-old patient with paraparesis of the lower extremities partially recovered after 10 weeks. She then walked without crutches, but her walk remained cumbersome.

### 2.3. Complications

Two psychiatric complications occurred shortly after hospital discharge. Two patients, previously untreated for psychiatric illnesses, experienced psychiatric complications in the late acute phase of TBE, after hospital discharge. One 71-year-old patient developed depression and one 48-year-old patient, bipolar disorder. Both needed acute hospitalization at a psychiatric department and subsequent long-term treatment. 

### 2.4. Laboratory Workup

A lumbar puncture was performed on 28 patients. Four patients disagreed with lumbar puncture; in three patients, the lumbar puncture was unsuccessful, and one patient was primarily admitted to another hospital and transferred to our department after confirmation of the diagnosis. The *cerebrospinal fluid* (CSF) was clear on appearance, in all cases, and in all CSF samples, leucocyte levels were elevated (>15 /mm^3^) (min 19, max > 1000). The leucocyte counts in CFS in two cases were more than one thousand and the precise level of total leucocytes and their types were not determined. Twenty-six of the twenty-eight patients tested had elevated protein levels (>0.45 g/L) in the CSF. The average protein level was 0.93 g/L (min 0.38, max 2.56 g/L). The diagnosis of TBE in eight patients without lumbar puncture was determined based on the positivity of both IgM and IgG-specific anti-TBEV antibodies in the serum. All of them had clinical symptoms of neuroinfection and positive epidemiological history.

#### 2.4.1. Blood Testing

The leukocytes count was mildly increased in most patients, with an average level of 13 × 10^9^/l (min 8, max 19 × 10^9^/L). C-reactive protein (CRP) had an average level of 19.6 (min < 5, max 76 mg/L), and the level of procalcitonin was normal in every patient (min 0.02, max 0.22 µg/L).

#### 2.4.2. Serology

The presence of specific anti-TBEV IgM antibodies in blood serum was confirmed in all 36 hospitalized patients, and in 29 of them, the IgG antibodies were positive as well. The level of IgG in three cases was equivocal, and in four cases, the specific anti-TBEV IgG antibodies in serum were negative. The sera samples were taken upon hospitalisation. Anti-TBEV-specific IgM antibodies were detected in the CSF in 14 of the 28 patients tested (2 with a negative serum IgG sample, 1 with equivocal and the others with positive IgG), and 11 of them had positive IgG antibodies in CSF as well, with one having IgG equivocal. Based on the definition criteria of the European Centre for Disease Control (ECDC) Meeting Report in 2011, TBE was confirmed in 32 cases (by detection of TBEV-specific IgM and IgG antibodies in the serum 29×, or by the IgM in the CSF–3×) and was probable in 4 patients (detection of TBEV-specific IgM antibodies in a unique serum sample) [13]. Neither blood nor CSF were tested for second sample specific anti-TBEV antibodies.

The results of CSF and blood analysis in all patients in the study group are attached in the Table A1 in Appendix A.

### 2.5. Long-Term Outcomes

After 3 years, 13 of the 27 patients (48%) who were contacted reported being well (6 men, 7 women) and had no persistent difficulties after overcoming TBE. The remaining 14 patients (52%—4 men, 10 women) reported that, compared to the pre-TBE period, they feel worse, and some of the following symptoms, listed in Table 1, persisted. Two patients had only one symptom, while others had two or more symptoms. One patient reported recurrent herpes eyelid infections that she did not have before contracting TBE.

### 2.6. Statistics

No statistically significant relationship was found between any of the tested variables and sex of the patients. We found that acute phase complications progress with age (Table 2).

The correlation between the presence of late symptoms (headache, memory disorders, sleep disorders, increased irritability, vertigo and psychiatric disorders), in relation to gender and the severity of acute TBE (meningitis/meningoencephalitis), was evaluated statistically, using Fisher’s exact test. No statistically significant relationship was found between any of the tested variables (Table 3).

The relationship between the occurrence of some symptoms (memory disorders, sleep disorders, vertigo) in the acute phase and after 3 years was also statistically evaluated, and again, no significant dependence was found in any of them (Table 4). Interestingly, more patients reported memory impairment after three years (7) than during the acute phase (2).

## 3. Discussion

Tick-borne encephalitis became notifiable at the EU level in 2012. Slovakia reports the highest occurrence of alimentary tick-borne encephalitis in Europe, with up to 17% of TBE cases caused by food transmission [14]. Even the first and greatest outbreak caused by the oral route occurred in Slovakia in 1951 (formerly Czechoslovakia). There were 660 cases of the disease that occurred after the ingestion of unpasteurized cow milk, which was mixed with contaminated goat milk [14]. During the 2012–2019 period, smaller, mostly family, outbreaks have been reported yearly, after the consumption of unpasteurised goat or sheep milk or products based on them [12]. The TBE outbreak in the district of Košice in 2016 was exceptional in the high number of cases. Goat milk and its products were the most probable transmission factor in previous alimentary TBE in Slovakia [14]. This is consistent with studies reported from other European countries. TBE is much less frequently transmitted through cow or sheep milk [15]. Based on data from affected patients, the TBE outbreak in the Košice region was linked to the consumption of cheese from unpasteurised sheep milk. The official report says that laboratory examinations, performed by the Regional Veterinary and Food Administration after the outbreak, on 2 pool samples of raw sheep milk and 52 samples of sheep serum did not confirm the presence of the TBE virus RNA, but antibodies were not examined [12]. As the samples of cheese or milk from the time when the outbreak began were not tested, because they were not available, the contamination with TBE virus cannot be excluded. The virus was probably not detected due to the relatively long time that has passed before the Investigation. The TBE virus can be detected in goat milk up to 18 days post-infection, and in sheep milk, 2–7 days post-infection, with peak viral loads occurring on day 2 [16,17,18].

In the observed population, the mean age of the hospitalized patients was 48 years. Only adults older than 19 years are hospitalized at the Department of Infectology and Travel Medicine (DITM) in Košice; children under the age of 19 are hospitalized at the Children’s hospital at the Infectious Disease Department. Upon consultation with the authorities of this department, it was found that only one 14 year-old-boy, with a mild meningitis form of TBE, was hospitalized there after the consumption of the cheese. Two other children (10 and 12 years old) of the affected families had subclinical TBE infection. This correlates with the literature that TBE is often subclinical in childhood [11]. Similarly, in a Czech study targeting the incidence of TBE in children, the highest incidence was in the oldest age group of 15–19 year olds [19]. Most cases occur in adults 20 to 50 years of age, with a male predominance [11]. However, our set does not correlate with the latter statement, since the number of women exceeded the number of men affected (20 vs. 16). A possible explanation may be alimentary route of transmission. Male predominance reflects occupational exposure in forestry and farming [11]. Similarly, in the neighbouring Czech Republic, during the period 1997–2008, there were 64 cases of food-borne TBE (0.9%) among all the TBE cases in that period, and the proportion of men and women was almost the same, with 33 men and 31 women affected [20].

Twenty patients (56%) from the observed population developed meningoencephalitis with upper limb tremor (13/36) as the most common neurological impairment. Bogovič et al. also observed meningoencephalitis as the predominant form of acute TBE illness, affecting 62.1% (445/717) of patients participating in their study [21]. Similarly, the results of a European Multicentre Study, which analysed a large multicentre cohort of 553 patients with confirmed TBE classified 37.3% of patients as having meningitis, 49.2% meningoencephalitis, 2.7% meningomyelitis, and 10.5% meningoencephalomyelitis [22].

In a self-evaluation of their health condition, 3 years after the acute infection, up to 14 of the 27 patients reported that their health was worse than before the disease. The most frequent complaint was headache, followed by memory impairment and then sleeping disorders and increased irritability. These results are similar to those of a Polish study, in which 20.6% of 1072 analysed patients reported symptoms during the follow-up, 1-month post-hospitalisation. The most frequent subjective complaints were headache, memory impairment, psychiatric disorders—most often depression and neurological disorder—upper limbs paresis. They also reported that subjective symptoms were more frequent after 1-month post-hospitalisation than during the hospitalisation, while objective neurological symptoms during the hospitalisation were more pronounced than after 1 month [23]. Similarly, the abovementioned study from Kohlmaier recorded persisting symptoms or signs without recovery expectation in 27% of 298 followed patients [22].

Psychiatric complications developed in two patients during the later acute stage of the disease—depression and bipolar affective disorder with the need for long-term treatment. Two patients had memory dysfunction and two patients had insomnia. After 3 years, anxiety was reported by two other patients. Sleep disorders were reported by another six patients, and memory impairment by seven patients, none of whom reported memory impairment during the acute phase of TBE. In a study by other Polish authors, the majority of patients experienced psychiatric problems during the course of TBE in the acute phase of the disease, as well as in the late phase—6 months after disease onset. The most common psychiatric manifestations were depressive disorders, and the authors recommend psychiatric evaluation after termination of acute symptomatology. More data will be needed to link TBE and the late psychiatric complication. Neurological symptoms during follow-up (hemiparesis, paraparesis, tremor of hands, convulsions of legs) were reported by five patients. All five patients who reported persistent neurological symptoms had overcome a more severe course of the disease with encephalitis [24]. Similarly, authors from Lithuania reported permanent CNS dysfunction after 1 year in 8.5% of 133 TBE cases. The risk of incomplete recovery was significantly higher among patients with the encephalitic form of TBE [25].

Ličková et al., in their review article, state five main differences between alimentary and tick-transmitted TBE: shorter incubation period, more common biphasic course, less severe disease, higher probability of recovery and occurrence in outbreaks in alimentary infection. Based on the results of our study, we can agree that alimentary TBE occurs in small outbreaks and has a more frequent biphasic course, which we observed in 69% [26]. However, in our group of patients, alimentary TBE did not have a mild course of the disease (56% had meningoencephalitis) and persisting symptoms were found in 52% of the studied patients, and incubation period was not short, but 14 days on average. Further studies are needed to assess the nature of alimentary TBE.

## 4. Materials and Methods

During the outbreak of tick-borne encephalitis, 44 people became ill and 36 patients were hospitalized at the Department of Infectology and Travel Medicine in Košice. Patients were admitted to the hospital in the period from 24 May 2016 to 16 June 2016.

Data from the medical documentation of 36 hospitalized patients were retrospectively analysed. The group consisted of 20 women and 16 men, with a mean age of 48 years (ranging from 21 to 71 years). The epidemiological links, clinical symptoms, basic laboratory and serological tests during hospitalization and late consequences were analysed in the study. The clinical presentation of the disease was classified as meningoencephalitis, defined as the presentation of neurological impairment or as meningitis with high fever, cephalea, positive CFS findings but no focal neurological deficits.

The diagnosis was made using the ELISA serology tests Enzygnost Anti-FSME/TBE-virus (Siemens Healthcare Diagnostic).

To determine the long-term symptoms and consequences of the disease we reached out to patients three years after the onset of the disease. The patients were contacted once by telephone.

Patients were contacted by telephone three years after the onset of the disease to determine the long-term symptoms and consequences of the disease. Out of the 36 patients, this information was obtained from 27 patients. In the meantime, one patient died of liver cancer; one patient refused to answer the questions, and the other 7 could not be contacted. Patients answered the question: Do you have persistent symptoms after overcoming TBE, such as headache, vertigo, sleep disorders, increased fatigue, irritation, impaired concentration or memory, or psychiatric symptoms such as depression and anxiety, neurological symptoms or others?

Associations between the sex of the TBE patients and three selected symptoms (severity of the acute infection e.g., meningitis/encephalitis, occurrence of acute phase complications and late consequences) were statistically evaluated using Pearson’s chi-squared test or Fisher’s exact test. Associations between age of the TBE patients and these three symptoms were statistically evaluated using logistic regression models. The presence/absence of late symptoms was statistically evaluated in correlation with the course of the acute phase of TBE using Fisher’s exact test.

## 5. Conclusions

The clinical spectrum of tick-borne encephalitis ranges from a mild, short-lived illness to a more severe, life-threatening illness. The disease can have serious long-term consequences, especially mental, neurologic and psychiatric disorders. In our group of patients with alimentary TBE, we observed common biphasic course (69%), relatively severe course of the disease and common consequences. Of the infected patients, 36 of 44 were hospitalized and 56% of hospitalised patients had meningoencephalitis, 44% had meningitis; 52% of respondents subjectively reported that, 3 years after TBE, their health was worse than before contracting the disease. The incidence of these late symptoms did not correlate with the severity of the acute phase of the disease. In our study, we also observed severe organ complications in the late acute phase of the disease; however, the direct association of these complications with TBE remains questionable and more data will be needed.

## Figures and Tables

**Table 1 pathogens-11-00433-t001:** The incidence of initial symptoms in the patient population and symptoms after 3 years.

Acute Symptoms	Number of Patients (Total 36)	Symptoms after 3 Years	Number of Patients (Total 27)
Fever	36 (100%)	Headache	7 (25%)
Headache	36 (100%)	Memory impairment	7 (25%)
Muscle weakness	31 (86%)	Sleeping disorders	6 (22%)
Postural instability	19 (53%)	Increased irritability	5 (19%)
Nuchal rigidity	18 (50%)	Vertigo	4 (15%)
Abdominal pain	17 (47%)	Concentration issues	3 (11%)
Vomiting	13 (36%)	Fatigue syndrome	2 (7%)
Upper limb tremor	13 (36%)	Dysarthria	1 (4%)
Confusion	6 (16.5%)	Learning difficulties	1 (4%)
Photophobia	6 (16.5%)	Psychiatric disorders	3 (11%)
Paresis	2 (5.5%)	Paresis	2 (7%)

**Table 2 pathogens-11-00433-t002:** Association between age of the TBE patients and three selected symptoms by means of logistic regression models.

Logistic Regression Models’parameters						
	Number of TBE Patients	Estimate	Std. Error	*p*-Value	Confidence Interval	
Encephalitis		0.0233	0.0271	0.3900	−0.0290	0.0795
Yes	20					
No	16					
Other complication		0.0842	0.0386	0.0293	0.0155	0.1705
Yes	9					
No	27					
Late symptoms		0.0260	0.0319	0.4150	−0.0353	0.0931
Yes	14					
No	13					

**Table 3 pathogens-11-00433-t003:** Association between the course of acute TBE infection and analysed late symptoms by means of Fisher’s exact test. (E—encephalitis; M—meningitis).

Severity of TBE	Total Analyzed Patients	With Particular Late Symptoms	Without Particular Late Symptoms	Fisher’s Exact Test *p* Value
Any late symptom				1
E	17	9	8	
M	10	5	5	
Headache				0.204
E	17	6	11	
M	10	1	9	
Memory disorders				0.204
E	17	6	11	
M	10	1	9	
Sleeping disorders				0.3625
E	17	5	12	
M	10	1	9	
Increased irritability				0.621
E	17	4	13	
M	10	1	9	
Vertigo				1
E	17	3	14	
M	10	1	9	
Psychiatric disorders				1
E	17	2	14	
M	10	1	9	

**Table 4 pathogens-11-00433-t004:** Association between the occurrence of particular symptoms at the time of acute TBE infection and 3 years later of Fisher’s exact test.

Symptom during the Acute TBE	Total Analyzed Patients	With Symptoms 3 Years Later	Without Symptoms3 Years Later	Fisher’s Exact Test *p* Value
Memory disorders				1
Yes	1	0	1	
No	26	7	19	
Sleep disorders				1
Yes	1	0	1	
No	26	6	20	
Vertigo				1
Yes	3	0	3	
No	24	4	20	
Gender		Male	Female	0.4401
Any symptom	14	4	10	
No symptom	13	6	7

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
