# Peer review of "Outbreak of Alimentary Tick-Borne Encephalitis in Eastern Slovakia: An Analysis of Affected Patients and Long-Term Outcomes"

_pathogens, 2022, doi:10.3390/pathogens11040433_

Round 1

Reviewer 1 Report

In this retrospective review of patients with TBE presumed to have been contracted by consuming cheese from infected sheep, the authors describe the spectrum of symptoms experienced by the 36 identified patients. Several aspects of the study limit its ability to shed light on differences between infections transmitted by an alimentary mechanism, vs those  due to tick-bite transmitted infection  - in particular, it is difficult to know how representative the data are. The study is retrospective, follow up information is missing in 25%, the conclusion that cases were due to cheese ingestion is inferred and not proven and it is not obvious whether this group constituted the entire population of patients with symptomatic infection.

That notwithstanding, the observations are interesting. Unfortunately, information about tick-bite associated infection is limited to previous reports in the literature. Would it be possible to look at tick-bite associated patients seen in the same institution over the same period of time, presumably providing a more comparable contrasting group?

Finally, although the follow-up information is intriguing, particularly in this age of ‘long haul COVID’ this appears to have been collected by asking a limited set of questions: i.e “Do you have persistent symptoms after overcoming TBE, such as headache, vertigo, sleep disorders, increased fatigue, irritation, impaired concentration or memory, or psychiatric symptoms such as depression and anxiety, neurological symptoms or others”, a method that biases responses towards these specific questions but loses any information about other possibilities

Author Response

Thank you for your detailed review of our article and useful comments and suggestions for improvement.

Point 1: That notwithstanding, the observations are interesting. Unfortunately, information about tick-bite associated infection is limited to previous reports in the literature. Would it be possible to look at tick-bite associated patients seen in the same institution over the same period of time, presumably providing a more comparable contrasting group?

Response 1: According to your question, we have checked the tick-bite associated patients seen in the same institution over the same period of time. In mentioned year, there were only two other patients with TBE hospitalized in our department and only one case was associated with tick-bite. So we are not able to provide contrasting group. Also, the TBE in our region is not so common, we usually have up to 6 patients during a year, that is why we believe this out-break is so interesting.

Point 2: Finally, although the follow-up information is intriguing, particularly in this age of ‘long haul COVID’ this appears to have been collected by asking a limited set of questions: i.e “Do you have persistent symptoms after overcoming TBE, such as headache, vertigo, sleep disorders, increased fatigue, irritation, impaired concentration or memory, or psychiatric symptoms such as depression and anxiety, neurological symptoms or others”, a method that biases responses towards these specific questions but loses any information about other possibilities.

Response 2: The outbreak occurred in 2016, and data were collected 3 years later in summer 2019, so we believe there is no connection of long haul COVID.

Reviewer 2 Report

Dear authors,

Congratulations for conducted study of alimentary tick-borne encephalitis in Košice region. 

In my opinion, the report is sound, but I have several comments and suggestions to be addressed by authors:

  1. Results section - Complications: I really dont understand how authors managed to link interacerebral haemorrhage, pulmonary embolism, and rupture of diaphragm as complications of TBE. We dont see here any anamnestic data, comorbidities or medical history of these patients that could give more probable cause of this pathological entities. Same goes with depression and bipolar disorder. You need to give more data if you want to link TBEV infection as cause of these disorders. I assume that nuchal rigidity is referred to positive meningeal sings? If that is true, please specify which meningeal findings were positive.
  2. Laboratory workup: Why CSF has not been analyzed in all patients? How did you come to the final diagnosis of CNS infection without CSF analysis? Please write down values of all CSF and blood laboratory analyzes, you can decide if you want to put that in main body of manuscript as table or as a supplementary file. These values could be actually important to other MDs dealing with alimentary TBE.
  3. Serology:  you searched for antibodies reactive with virus, not the disease, therefore you should write anti-TBEV instead anti-TBE. Please correct that in the whole manuscript. Why serology was not checked in all hospitalized patients? Why confirmation serology was not performed in seroreactive samples (WB, neutralization assay)?
  4. Table 4 - What criteria did you use to differentiate memory disorders and sleeping disorder from psychiatric disorders? As I am aware, both memory disorders (qualitative and quantitative) and sleeping disorder (insomnia) are psychiatric disorders also.
  5. Discussion: Line 176 - What does (SR) mean?
  6. Please rephrase conclusions accordingly to your responds related to my questions/comments. Especially considering complications.  

Author Response

Thank you for your detailed review of our article and useful comments and suggestions for improvement.

Point 1: Results section - Complications: I really dont understand how authors managed to link interacerebral haemorrhage, pulmonary embolism, and rupture of diaphragm as complications of TBE. We dont see here any anamnestic data, comorbidities or medical history of these patients that could give more probable cause of this pathological entities. Same goes with depression and bipolar disorder. You need to give more data if you want to link TBEV infection as cause of these disorders.

Response 1: We agree that direct connection between mentioned complication and TBE is questionable. We have added a sentence in the article in part „Complication “, line 98-99.: “Although these health complications have occurred, the direct link to TBE is questionable.” Also we mentioned this context in part „Conclusion “.

Point 2: I assume that nuchal rigidity is referred to positive meningeal sings? If that is true, please specify which meningeal findings were positive.

Response 2: We have added explanation of meningeal symptoms. Line 70-71.  “The typical meningeal symptoms are headache, fever, nuchal rigidity, postural instability, nausea and vomiting, photophobia and confusion.”

Point 3: Why CSF has not been analyzed in all patients?

Response 3: In laboratory workup we have added explanation why lumbar puncture in all patients have not been analyzed. „Four patients disagreed with lumbar puncture, in 3 patients the lumbar puncture was unsuccessful, and one patient was primarily admitted to another hospital and transferred to our department after confirmation of the diagnosis.” Line 105-107.

Point 4: How did you come to the final diagnosis of CNS infection without CSF analysis?

Response 4: The serology blood testing was performed in all hospitalized patients. In patients without lumbar puncture the diagnosis was confirmed on the basis of typical clinical signs, epidemiological context and positive serology test. Confirmatory tests e.g. WB were not available in our laboratory in this time.

Point 5: Please write down values of all CSF and blood laboratory analyzes, you can decide if you want to put that in main body of manuscript as table or as a supplementary file. These values could be actually important to other MDs dealing with alimentary TBE.

Response 5: Detailed presentation of all results would be very opaque that reason we presented statistical overview of attribute.

Point 6: You searched for antibodies reactive with virus, not the disease, therefore you should write anti-TBEV instead anti-TBE. Please correct that in the whole manuscript.

Response 6: We have corrected all names of antibodies to your suggested form.

Point 7: Why serology was not checked in all hospitalized patients?

Response 7: The serology blood testing was performed in all hospitalized patients.

Point 8: Why confirmation serology was not performed in seroreactive samples (WB, neutralization assay)?

Response 8: Confirmatory tests e.g. WB were not available in our laboratory in this time.

Point 9: What criteria did you use to differentiate memory disorders and sleeping disorder from psychiatric disorders? As I am aware, both memory disorders (qualitative and quantitative) and sleeping disorder (insomnia) are psychiatric disorders also.

Response 9: We believe that psychiatric disorders and TBE is connected, we discuss it in the part „Discussion “, but we have added a sentence, that more data will be needed. Line 248-249.6. We have corrected all names of antibodies to your suggested form. We used subjective evaluation of the patients.

Point 10: What does (SR) mean?

Response 10: It means Slovak republic, but we have deleted (SR).

Point 11: Please rephrase conclusions accordingly to your responds related to my questions/comments. Especially considering complications. 

Response 11: We have corrected the last sentence in “Conclusion” according to your comments.

Round 2

Reviewer 1 Report

This interesting report of milk-transmitted TBE continues to have significant issues.  My biggest concern is in labelling post-infectious difficulties such as irritability, sleep disturbance, headaches, subjective memory difficulties, depression and bipolar disease as sequelae – particularly since the implication is they were due to neurologic damage, when in fact, they were not associated with encephalitis.

A few specific suggestions:

  1. Define encephalitis: in patients with a demonstrated CSF pleocytosis and either focal abnormalities on neurologic exam or on MRI, this is a plausible diagnosis. In those with confusion, photophobia, sleeping disorders, irritability – not. Even altered consciousness can be a non-specific response to systemic infections. Please indicate in Table 1 which symptoms you feel are indicative of encephalitis.
  2. Post treatment symptoms. My allusion to COVID long-haulers was not to suggest these patients had COVID but rather that long-haulers have led to a focus on post-infectious symptoms.  Although it is interesting to list these here, absent information on their frequency in control populations (either after other infections or in healthy normals) interpretation must be EXTREMELY cautious
  3. Patients without lumbar punctures: Please list the “typical clinical signs” in these patients, and which of them were thought to have meningitis vs meningoencephalitis (could be in Table 1)
  4. I do not think Table 2 adds anything useful

Author Response

Thank you for your review and useful comments.

  • We didn´t intend to discuss encephalitis in the table 1. It´s part of the following text. 
  • We added Limitations of the study in which we discuss this topic.
  • We corrected that in the text.
  • We removed table 2 based on your recommendation.

Reviewer 2 Report

Authors made some corrections, but the following points still need to improve. 

  1. you didn't add laboratory data in supplementary file
  2. you didn't perform confirmation serology analysis 
  3. The diagnosis of CNS infection without lumbar puncture in described patients is highly questionable 

Author Response

Thank you for your review and useful comments.

  • We added the data as you requested in supplementary file.
  • The outbreak happened 5 years ago, we didn´t have confirmation serology analysis neither PCR available at the time.
  • As we added in the text, we set the diagnosis based on the positive serology test and clinical symptoms of neuroinfection.

Round 3

Reviewer 1 Report

Thank you for your responses